# MicroRNA-148a Regulates the Proliferation and Differentiation of Ovine Preadipocytes by Targeting *PTEN*

**DOI:** 10.3390/ani11030820

**Published:** 2021-03-15

**Authors:** Xiayang Jin, Zhiyun Hao, Mengli Zhao, Jiyuan Shen, Na Ke, Yize Song, Lirong Qiao, Yujie Lu, Liyan Hu, Xinmiao Wu, Jiqing Wang, Yuzhu Luo

**Affiliations:** Gansu Key Laboratory of Herbivorous Animal Biotechnology, Faculty of Animal Science and Technology, Gansu Agricultural University, Lanzhou 730070, China; jinxiayang2018@163.com (X.J.); haozy2018@163.com (Z.H.); 18394187234@163.com (M.Z.); shenjy@st.gsau.edu.cn (J.S.); ken@st.gsau.edu.cn (N.K.); songyz@st.gsau.edu.cn (Y.S.); qiaolr735757@163.com (L.Q.); luyujie1113@163.com (Y.L.); huliyan2020@163.com (L.H.); wuxinmiao2020@163.com (X.W.)

**Keywords:** miR-148a, proliferation, adipogenesis, ovine preadipocytes, *PTEN*

## Abstract

**Simple Summary:**

The miR-148a has been shown to play an important role in preadipocyte differentiation. Herein, we explored the role of miR-148a in ovine adipocyte development, using Oil Red O staining, CCK-8, EdU, flow cytometry and RT-qPCR. The results showed that miR-148a suppressed proliferation and facilitated the differentiation of preadipocytes. The dual fluorescent reporter vector experiments showed that miR-148a directly targeted *PTEN*. Meanwhile, we demonstrated that *PTEN* significantly inhibited the differentiation of preadipocytes. In conclusion, our research provides new insights that miR-148a inhibits ovine preadipocyte proliferation and accelerates differentiation by the negative regulation of *PTEN*.

**Abstract:**

MicroRNAs (miRNAs) have been found to be involved in lipid deposition and metabolism. However, there have been no reports on the roles of miR-148a in the proliferation and adipogenesis of preadipocytes in sheep. In this study, the expression of miR-148a was profiled in the eight tissues of Tibetan ewes and differentiated preadipocytes, and the role of miR-148a in differentiation and proliferation of ovine preadipocytes was investigated using Oil Red O staining, CCK-8, EdU staining, cell cycle detection, and RT-qPCR. The effect of *PTEN* on the differentiation of ovine preadipocytes was also investigated. The miR-148a was widely expressed in the eight tissues investigated and had significantly increased expression in liver, spleen and subcutaneous adipose tissues, and the heart. The expression of miR-148a continued to increase with the differentiation of ovine preadipocytes. The over-expression of miR-148a significantly promoted differentiation but inhibited the proliferation of ovine preadipocytes. The inhibition of miR-148a had the opposite effect on the differentiation and proliferation of ovine preadipocytes with over-expressed miR-148a. The results from the dual luciferase reporter assays showed that miR-148a mimic significantly decreased the luciferase activity of *PTEN*-3′UTR dual luciferase reporter vector, suggesting that *PTEN* is a target gene of miR-148a. In over-expressed-*PTEN* preadipocytes, the number of lipid droplets remarkably decreased, and the expression levels of adipogenesis marker genes *PPARγ*, *FASN*, *FATP4*, *GLUT4*, *C/EBPβ* and *LPL* were also significantly down-regulated. These results suggest that miR-148a accelerated the adipogenic differentiation of ovine preadipocytes by inhibiting *PTEN* expression, and also inhibited the proliferation of ovine preadipocytes.

## 1. Introduction

Adipose tissue plays important roles in providing energy, maintaining body temperature, and protecting internal organs in animal bodies [1]. The fat deposition also affects the production performance of livestock, especially meat quality [2]. It has been reported that moderate fat deposition improves the tenderness and flavor of mutton in sheep [3]; thus, an in-depth understanding of the molecular mechanisms that regulate fat tissue development provides the opportunity to improve meat quality in sheep. The development of adipose tissue has been found to be closely associated with the proliferation and differentiation of preadipocytes [4]. It is well-known that the proliferation and differentiation of preadipocytes are regulated by functional genes and non-coding RNAs [5,6].

MicroRNAs (miRNAs) are a class of evolutionarily conserved non-coding RNAs. They are widely involved in the regulation of a series of biological processes and cell activities at the post-transcriptional level through mRNA cleavage or translation inhibition, including cell fate, proliferation, differentiation, apoptosis, and tumorigenesis [7,8]. It has been found that miRNAs are involved in the proliferation and adipogenesis of preadipocytes. For example, Li et al. [9] found that the over-expression of miRNA-223 suppressed the differentiation of chicken intramuscular preadipocytes by decreasing the expression level of *GPAM*. The up-regulation of miR-149-5p inhibited the proliferation and differentiation of bovine preadipocytes by negatively regulating the expression of *CRTCs* [10]. The over-expression of miRNA-125a-5p has been found to promote the proliferation of preadipocytes but inhibit their differentiation in porcine preadipocytes [11]. The function of other miRNAs has also been elucidated in adipogenesis, including miR-204-5p [12], miR-145 [13], miR-20a-5p [14], miR-106a [15], and miR-127 [16].

To date, the function of miR-148a has mainly focused on tumorigenesis [17,18,19]. In some mammals, it has also been confirmed that miR-148a plays an important role in adipogenesis. Shi et al. [20] reported that the expression of miR-148a was up-regulated in hMSCs-Ad cells and promoted adipogenesis by inhibiting the expression of *WNT1*. The promotion of miR-148a on preadipocytes has also been reported in rabbits [21]. The over-expression of miR-148a promoted the synthesis of triglycerides by down-regulating the expression of *PPARGC1A* and *PPARA* in goat mammary epithelial cells [22]. However, there have been no reports on the effect of miR-148a on the proliferation, adipogenesis, and lipid deposition of preadipocytes in sheep.

In this study, we investigated the expression profile of miR-148a and its effect on the proliferation and adipogenesis of ovine preadipocytes. We also detected the target gene of miR-148a and further evaluated the regulatory effect of miR-148a on the target gene.

## 2. Materials and Methods

### 2.1. Ethics Statement

The experimental work followed the guidelines for the care and use of laboratory animals (approval number 2006-398; the Ministry of Science and Technology of China), and also approved by the Gansu Agricultural University.

### 2.2. Collection of Ovine Tissue Samples

A total of four healthy, eighteen-month-old Tibetan ewes were selected for investigation. All these ewes were raised under the same environmental conditions in Gannan Tibetan Autonomous Prefecture, China. The ewes were slaughtered to collect eight tissue samples for subsequent reverse transcription quantitative PCR (RT-qPCR) analysis, including subcutaneous fat, *longissimus dorsi* muscle, testis, kidney, lung, spleen, heart, and liver tissues. These samples were immediately frozen in liquid nitrogen and then stored at −80 °C. Blood samples from these ewes were also collected and genomic DNA was then extracted using an EasyPure^®^ Blood genomic deoxyribonucleic acid Kit (TransGen Biotech, Beijing, China). Meanwhile, a part of the subcutaneous fat tissue was collected and then placed in PBS containing 10% penicillin/streptomycin antibiotics (Hyclone, Logan, UT, USA) for culturing ovine preadipocytes.

### 2.3. Isolation of Ovine Primary Preadipocytes and Cell Culture

After removing visible blood vessels and connective tissues, the samples were cut to 0.5 to 1.0 mm^3^ pieces and then digested using buffers with 0.75 U/mL collagenase D (Solarbio, Beijing, China) and 1.0 U/mL Dispase type II (Solarbio, Beijing, China) at 37 °C for 1 h. The digestive fluid was filtered and the preadipocytes were collected. After being washed three times with PBS, the ovine preadipocytes were precipitated and then resuspended using DMEM-F/12 medium (Hyclone, Logan, UT, USA) containing 10% fetal bovine serum (Invigentech, Xi’an, China). Finally, the preadipocytes were uniformly inoculated in the 60 mm plate and then cultured at 37 °C and 5% CO_2_ for 24 h.

### 2.4. RNA Isolation and RT-qPCR Analysis

TRIzol (Vazyme, Nanjing, China) was used to extract the total RNA of ovine adipocytes and the eight tissues collected. The cDNAs were synthesized using a HiScript III 1st Strand cDNA Synthesis Kit (Vazyme, Nanjing, China). *U6* [23] and *TBP* [24] were chosen as internal references to normalize the expression of miRNAs and mRNAs, respectively. The RT-qPCR was performed in triplicate using the 2× ChamQ SYBR qPCR Master system (Vazyme, Nanjing, China) on an Applied Biosystems QuantStudio 6 Flex Real-time PCR System (Thermo Fisher Scientific, Waltham, MA, USA). The information of PCR primers is listed in Appendix A. The relative expression level of the RNA was calculated using a 2^−ΔΔCt^ method.

### 2.5. Preadipocytes Differentiation and the Staining of Lipid Droplets

According to the method suggested by Ma et al. [25], the inducers of adipogenic differentiation (27.8 μg/mL 3-isobutyl-1-methylxanthine, 0.1 μg/mL dexamethasone, and 1 μg/mL insulin) were added to the original growth medium to induce the differentiation of preadipocytes for 2 days. The induction medium was then replaced by a maintenance medium (growth medium supplemented with 1 μg/mL insulin) and adipocytes were further cultured for 2 days. The maintenance medium was substituted with original growth medium, and the culture of adipocytes continued until the differentiation process was finished. The whole differentiation process lasted for 8 days, and mature ovine adipocytes were obtained. For investigating the expression profiles of miR-148a and the target gene *PTEN* in the lipogenesis phase of ovine adipocytes, RNA was extracted from adipocytes on day 0, 0.5, 1, 2, 4, 6, and 8 after differentiation began. The expression of adipogenic marker gene fatty acid binding protein 4 (*aP2*) was also detected using RT-qPCR.

The miR-148a mimic, miR-148a inhibitor, and non-specific control (NC) for mimic and inhibitor were synthesized by RiboBio Ltd. (Guangzhou, China). When the density of preadipocytes attained approximately 90%, the preadipocytes were transiently transfected with miR-148a mimic, miR-148a inhibitor and two groups of NC. The same inducers and procedure described above were used to induce the differentiation of ovine preadipocytes. On day 8 after differentiation, Oil Red O staining and RT-qPCR of lipogenesis markers were performed.

For Oil Red O staining, the mature adipocytes were fixed in 4% paraformaldehyde for 30 min and then washed by PBS 3 times, followed by being stained with 1% filtered Oil Red O staining solution for 20 min. An IX53 inverted fluorescence microscope (Olympus, Tokyo, Japan) was used to observe the stained lipid droplets in the cytoplasm of adipocytes. Meanwhile, the expression level of lipogenesis markers *aP2*, peroxisome proliferator activated receptor gamma (*PPARγ*), fatty acid synthase (*FASN*), solute carrier family 2 member 4 (*GLUT4*) and lipoprotein lipase (*LPL*) were detected.

### 2.6. Preadipocyte Proliferation Analysis

To investigate the effect of miR-148a on ovine preadipocyte proliferation, when the density of preadipocytes achieved approximately 50%, ovine preadipocytes were plated into 96-well plates and then transfected using miR-148a mimic, miR-148a inhibitor and two groups of NC. First, the cell viability of preadipocytes was detected at 0, 6, 12, 24, and 48 h after transfection using a CCK-8 Cell Counting Kit (Vazyme, Nanjing, China). To each well was added 10 μL CCK-8 solution which was then cultured at 37 °C for 2 h. The absorbance was measured at 450 nm wavelength using a Varioskan LUX Multimode Reader (Thermo Fisher Scientific, MA, USA). Secondly, the DNA synthesis rate was detected at 48 h after transfection to reflect the rate of preadipocyte proliferation using the BeyoClick™ EdU-488 kit (Beyotime, Shanghai, China). An IX53 inverted fluorescence microscope (Olympus, Tokyo, Japan) was used to observe fluorescence from the stained nucleus. From each group of preadipocytes, four images were randomly collected for statistical analysis. Thirdly, the expression levels of proliferation marker genes cyclin-dependent kinase 2 (*CDK2*), cyclin-dependent kinase 4 (*CDK4*), CyclinB1, proliferating cell nuclear antigen (*PCNA*) and *p53* were detected using RT-qPCR to explore the effect of miR-148a on sheep preadipocyte proliferation on day 2 after transfection. Finally, a cell cycle analysis kit (Thermo Fisher Scientific, Waltham, MA, USA) and AccuriC6 flow cytometry system (BD Biosciences, New Jersey, NJ, USA) were used to detect the content of DNA in each cell cycle of preadipocytes.

### 2.7. The Prediction of Target Genes of miR-148a

The target genes of miR-148a were predicted using TargetScan and miRDB. The predicted results from the two kinds of software were intersected. Of a total of four hundred and thirty target genes predicted, *PTEN* was selected as a candidate gene to verify the target relationship with miR-148a, according to the prediction score and previous results in glomerular cell of mice [26].

### 2.8. Dual Luciferase Reporter Assay

For constructing the wild-type *PTEN*-3′UTR dual luciferase reporter vectors, the primers were designed to amplify the *PTEN*-3′UTR sequences, which contained the seed site of miR-148a.The 3′UTR sequence of *PTEN* was connected to the 3′-end of the Ranilla luciferase reporter gene of the pmiR-RB-Report™ vector (RiboBio, Guangzhou, China) and then the wild-type dual luciferase reporter vectors were obtained. Meanwhile, a Mut Express II Fast Mutagenesis Kit (Vazyme, Nanjing, China) was applied to construct *PTEN*-3′UTR mutant-type reporter vectors.

The wild-type or mutant-type *PTEN*-3′UTR dual luciferase reporter vectors (500 ng) and miR-148a mimic (100 pmoL) or NC (100 pmoL) were co-transfected into HEK293 cells. The HEK293 cells were collected after 48 h and the luciferase activity was detected by the dual luciferase reporter assay system (Promega, Madison, WI, USA).

### 2.9. The Effect of PTEN on Differentiation of Ovine Preadipocytes

The small interfering RNA of *PTEN* (si-*PTEN*) was synthesized by GenePharma Ltd. (Shanghai, China). The synthesized RNA was diluted to a final concentration of 50 nM. The CDS region of *PTEN* was amplified using the cDNA template and then ligated into the pcDNA3.1 vector (Invitrogen, Carlsbad, CA, USA) to construct an expression vector (named pcDNA3.1-*PTEN*).

When the density of preadipocytes reached approximately 90%, si-*PTEN*, 2 μg pcDNA3.1-*PTEN* plasmid, 4 μg pcDNA3.1-*PTEN* plasmid, and the empty pcDNA3.1 plasmid were transfected into respective ovine preadipocytes using INVI DNA RNA Transfection ReagentTM (Invigentech, Xi’an, China). The same inducers and procedure described above were used to induce the differentiation of ovine preadipocytes. On day 8 after differentiation, the expression levels of adipogenesis marker genes *PPARγ*, *FASN*, solute carrier family 27 member 4 (*FATP4*), *GLUT4*, CCAAT/enhancer binding protein beta (*C/EBPβ*) and *LPL* were detected. The number of lipid droplets was also analyzed using Oil Red O staining.

### 2.10. Statistical Analysis

All data were analyzed using SPSS 22.0 software and the data are presented as mean ± SD for three replicates. Statistical significance was determined using one-way analysis of variance. All *p*-values were considered statistically significant when *p* < 0.05.

## 3. Results

### 3.1. The Expression Profile of miR-148a

It was deduced from the RT-qPCR analysis in the eight different tissues of Tibetan ewes that miR-148a was robustly expressed in liver, spleen, and subcutaneous adipose tissues and the heart, while it had a weak expression in kidney, lung, longissimus dorsi muscle and testis (Figure 1A).

Fatty acid binding protein 4 (*aP2*) is a marker gene of adipogenic differentiation that reflects whether adipocytes normally differentiate. The expression level of *aP2* was remarkably increased on day 2 and reached the maximum on day 6 after preadipocyte differentiation, and then significantly decreased in the late stage of differentiation (Figure 1B). This confirmed that the preadipocytes were normally differentiated. The expression level of miR-148a continuously increased from 0 day to 8 day of adipogenic differentiation (Figure 1C).

### 3.2. The Positive Regulation of miR-148a in the Differentiation of Ovine Preadipocytes

To evaluate the biological role of miR-148a in the adipogenesis of ovine preadipocytes, miR-148a mimic, miR-148a inhibitor, and two groups of NC were transfected into preadipocytes. The content of lipid droplets and the expression levels of lipogenesis marker genes were then detected in adipocytes on day 8 after adipogenic differentiation using Oil Red O staining and RT-qPCR, respectively. Compared to the miR-148a mimic NC group, the expression level of miR-148a remarkably increased when preadipocytes were transfected with miR-148a mimic (Figure 2A,B). On the contrary, compared to the miR-148a inhibitor NC group, the expression level of miR-148a remarkably decreased when the miR-148a inhibitor was transfected into preadipocytes (Figure 2A,B). This suggests that the miR-148a mimic, miR-148a inhibitor and NC were successfully transfected into ovine preadipocytes.

The over-expression of miR-148a significantly increased the accumulation of lipid droplets in the adipocyte cytoplasm (Figure 2C), while inhibited miR-148a decreased the accumulation of lipid droplets using Oil Red O staining (Figure 2E). The number of lipid droplets in the image was counted and the results were also consistent with Oil Red O staining (Figure 2D,F).

The RT-qPCR results of lipogenesis marker genes in adipocytes on day 8 after adipogenic differentiation showed that the transfection of miR-148a mimic significantly enhanced the expression levels of adipogenic marker genes *aP2*, *PPARγ*, *FASN*, *GLUT4* and *LPL*, while the expression levels of the genes were all decreased when the miR-148a inhibitor was transfected into ovine preadipocytes (Figure 3). These results suggest that miR-148a promoted the adipogenic ability of ovine preadipocytes.

### 3.3. MiR-148a Inhibited Ovine Preadipocyte Proliferation

It was shown that the preadipocytes were successfully transfected with miR-148a mimic, miR-148a inhibitor and NC (Figure 4A). The results of cell viability detections demonstrated that miR-148a had no significant effect on the proliferation activity of ovine preadipocytes at 0–24 h when miR-148a was over-expressed in preadipocytes. However, the over-expression of miR-148a markedly impaired the proliferation ability of ovine preadipocytes at 48 h after transfection. On the contrary, the inhibition of miR-148a increased the proliferation activity of ovine preadipocytes (Figure 4B).

EdU staining analysis was used to detect the efficiency of DNA synthesis in ovine preadipocytes during mitosis. The results revealed that the over-expression of miR-148a inhibited the birth of new preadipocytes, and the miR-148a inhibitor presented the opposite effect with the miR-148a mimic (Figure 4C,D).

The RT-qPCR results of proliferation marker genes showed that the expression levels of *CDK2*, *CDK4*, *CyclinB1* and *PCNA* were down-regulated, while *p53* was up-regulated in the over-expressed miR-148a preadipocytes when compared to the NC group (Figure 5). The transfection of miR-148a mimic decreased the proportion of preadipocytes in the S phase (Figure 6A,B). On the contrary, miR-148a inhibitor promoted the cell cycle moving of preadipocytes from G1/G0 phase to S phase (Figure 6C,D). The results were also supported by the statistical results of flow cytometry data (Figure 6E,F).

### 3.4. MiR-148a Targets the 3′UTR Region of PTEN

Dual luciferase reporter assays were used to verify the target relationship between miR-148a and *PTEN*. The structures of *PTEN*-3′UTR wild-type and mutant-type reporter vectors are shown in Figure 7A. When the miR-148a mimic and wild-type *PTEN*-3′UTR reporter vector were co-transfected into the HEK293 cells, the luciferase activity was significantly lower than in the HEK293 cells which were co-transfected with the mimic-NC and wild-type *PTEN*-3′UTR reporter vector. Meanwhile, when miR-148a mimic or mimic-NC and mutant-type *PTEN*-3′UTR reporter vectors were co-transfected into the HEK293 cells, there was no significant difference in luciferase activity between the group of miR-148a and mimic-NC (Figure 7B). This suggests that *PTEN* is a target gene of miR-148a.

The further results showed that the miR-148a mimic significantly decreased the expression level of *PTEN*, while the expression level of *PTEN* remarkably increased when the miR-148a inhibitor was transfected into preadipocytes (Figure 7C). In addition, the expression of *PTEN* gradually decreased with the differentiation of ovine preadipocytes (Figure 7D).

### 3.5. PTEN Inhibits Differentiation of Ovine Preadipocytes

When 2 μg or 4 μg pcDNA3.1-*PTEN* expression plasmid were transfected into ovine preadipocytes, the number of lipid droplets remarkably decreased (Figure 8A,B). The RT-qPCR results revealed that the expression level of *PTEN* was significantly enhanced in adipocytes transfected with pcDNA3.1-*PTEN* expression plasmid (Figure 8C). In addition, the expression levels of adipogenesis marker genes *PPARγ*, *FASN* and *FATP4* were significantly down-regulated, while the expression levels of *GLUT4*, *C/EBPβ* and *LPL* were not significantly changed when 2 μg pcDNA3.1-*PTEN* expression plasmid were transfected into preadipocytes (Figure 8D). However, when 4 μg pcDNA3.1-*PTEN* expression plasmid were transfected into preadipocytes, the expression level of adipogenesis marker genes was all significantly down-regulated (Figure 8D).

On the contrary, when *PTEN* was silenced in ovine adipocytes using siRNA, the accumulation of lipid droplets in adipocytes significantly increased (Figure 9A,B) and the expression level of *PTEN* was significantly decreased (Figure 9C). In addition, the expression levels of *PPARγ*, *FASN*, *GLUT4*, *C/EBPβ*, *FATP4*, and *LPL* were all significantly up-regulated when *PTEN* was silenced in ovine adipocytes (Figure 9D).

## 4. Discussion

In the last few years, many studies have confirmed that miRNAs play essential roles in the proliferation and adipogenesis of preadipocytes [10,23]. However, this is the first study to investigate the expression profiles of miR-148a, and roles of miR-148a in the proliferation and differentiation of ovine preadipocytes.

In this study, miR-148a was not only expressed in the eight tissues studied, but also had a higher expression level in liver, spleen, and subcutaneous adipose tissues and the heart than in the kidney, lung, *longissimus dorsi* muscle, and testis. This was consistent with findings in buffalo, in which the expression of miR-148a was significantly higher in subcutaneous adipose tissue than in *longissimus dorsi* muscle [27]. The higher expression of miR-148a in adipose tissue suggests that the miRNA may play an important role in regulating adipogenesis. Meanwhile, the expression of miR-148a gradually increased with the differentiation and maturation of ovine adipocytes. Londono et al. [28] also found similar results in human adipose-derived mesenchymal stem cells. This indicates that miR-148a may promote adipocyte differentiation and lipid accumulation in sheep. The conclusion was further confirmed by Oil Red O staining and RT-qPCR analysis of adipogenic differentiation marker genes. The over-expression of miR-148a promoted the accumulation of lipid droplets and increased the expression levels of *aP2*, *PPARγ*, *FASN*, *GLUT4* and *LPL*. *aP2* is mainly expressed in adipose tissue and mature adipocytes and regulates fatty acid uptake and intracellular transport in adipocytes [29]. The loss-of-function studies have demonstrated that *PPARγ* is necessary for the adipogenic differentiation of adipocytes. Rosen et al. [30] found that the knock-out of *PPARγ* significantly inhibited the development of adipose tissue in mice. Embryonic fibroblasts derived from *PPARγ*-deficient fetuses were unable to differentiate into adipocytes [31]. *FASN* encodes a rate-limiting enzyme for the de novo synthesis of long-chain fatty acids in the adipocytes of mammals. In mouse adipocytes, up-regulation of *FASN* expression significantly promoted the accumulation of intracellular triglycerides [32]. Increased expression of *CLUT4* improved insulin resistance and lipid accumulation in mouse adipose tissue [33]. *LPL* is an important gene for fat tissue and adipocytes to metabolize fatty acids, which promotes fatty acids in the blood to enter adipocytes to participate in the synthesis of triglycerides [34,35]. Cho et al. [36] also found that the over-expressed miR-148a increased the expression level of *PPARγ* and accelerated adipogenesis in mouse 3T3-L1 cells.

This is the first study to report that miR-148a inhibits the proliferation of ovine preadipocytes in sheep using CCK-8 and EdU staining. The inhibition of miR-148a on the proliferation of other types of cells was also reported. For example, Xu et al. [37] found that miR-148a suppressed human glioblastoma cells proliferation by targeting integrin subunit alpha 9 (*ITGA9*). The miR-148a inhibited the proliferation of skeletal muscle cells by down-regulating the expression level of *KLF6* [38]. Lv et al. [39] have also reported the inhibition of miR-148a on the proliferation of dermal papilla cells in Hu sheep. The percentage of preadipocytes in each cell cycle stage can also reflect the proliferation activity of the preadipocytes. It was found that the number of cells in the S phase is positively correlated with the proliferation activity of the cells [40]. Our results indicated that the number of preadipocytes in the S phase significantly decreased when miR-148a mimic was transfected into preadipocytes. Song et al. [38] found that miR-148a inhibited cells moving from mitotic G1 to the S phase in bovine skeletal muscle cells. The effect of miR-148a on the proliferation of preadipocytes was also reflected by the expression levels of proliferation marker genes. The over-expression of miR-148a inhibited the expression of *CDK2*, *CDK4*, *cyclinB1* and *PCNA* in preadipocytes, but promoted the expression of *p53*. *CDK2*, *CDK4*, *cyclinB1* and *PCNA* positively regulated cell proliferation by promoting the cell cycle from G1 to the S phase [41], while *p53* negatively regulated cell proliferation by inhibiting the expression of *CDK2* [42].

It was confirmed in the study that miR-148a directly targeted *PTEN* and then inhibited the expression of *PTEN* in ovine preadipocytes. Zhang et al. [43] also found that miR-148a can target *PTEN* in MG63 cells and inhibit the expression of *PTEN*. In addition, there was an opposite tendency in the study between the expression levels of miRNA-148a and *PTEN* during the differentiation of ovine preadipocytes. Specifically, the expression of miR-148a continuously increased, but the expression of *PTEN* gradually decreased. This result also supported the target relationship between miR-148a with *PTEN*. Zhang et al. [23] also found that the expression profile of miR-143a-3p was opposed to its target gene *MAPK7* during the adipogenic differentiation of 3T3-L1 cells.

In this study, we over-expressed and silenced *PTEN* in ovine preadipocytes using the pcDNA3.1 expression plasmid and siRNA, respectively. It was found that *PTEN* inhibited the differentiation of ovine preadipocytes, using Oil Red O staining and RT-qPCR for adipogenic marker genes (*PPARγ*, *FASN*, *GLUT4*, *C/EBPβ*, *FATP4* and *LPL*). These genes have been reported to be related with the differentiation of adipogenesis. For example, *C/EBPβ* has been considered as the first induced transcription factors during adipogenesis and plays an important role in initiating the adipogenesis of preadipocytes. Adipogenesis was impaired when *C/EBPβ* was knocked out in mouse adipose tissue [44]. The absorption of fatty acids and the accumulation of intracellular lipids were significantly enhanced in mouse 3T3-L1 cells when the expression level of *FATP4* was up-regulated [45]. Xu et al. [46] found that silenced *PTEN* increased the number of Oil Red O-stained amniotic fluid mesenchymal stem cells and accelerated the adipogenic differentiation of AF-MSCs. Lee et al. [47] also found that the inhibition of *PTEN* expression by RNAi potentiated the phosphorylation of Akt and then accelerated adipogenesis of the 3T3-L1 cells. Given the studies described above and our findings, it was inferred that miR-148a promoted the adipogenic differentiation of ovine preadipocytes by targeting *PTEN*.

For all the above-mentioned reasons, identification of molecular factors responsible for adipogenic differentiation and proliferation of ovine preadipocytes could increase the practical usefulness and suitability of these cells for somatic cell cloning in sheep and other mammalian species as an alternative source of nuclear donors [48,49] in relation to fetal fibroblast cells [50,51], adult dermal fibroblast cells [52,53], and adult mesenchymal stem cells [54,55].

## 5. Conclusions

In summary, up-regulation of miR-148a expression accelerated the adipogenic differentiation of ovine preadipocytes through targeting *PTEN*. Additionally, miR-148a resulted in cell cycle arrest and proliferation suppression in ovine preadipocytes. Therefore, miR-148a was able to serve as a key regulator of adipogenesis in ovine preadipocytes.

## Figures and Tables

**Figure 1 animals-11-00820-f001:**
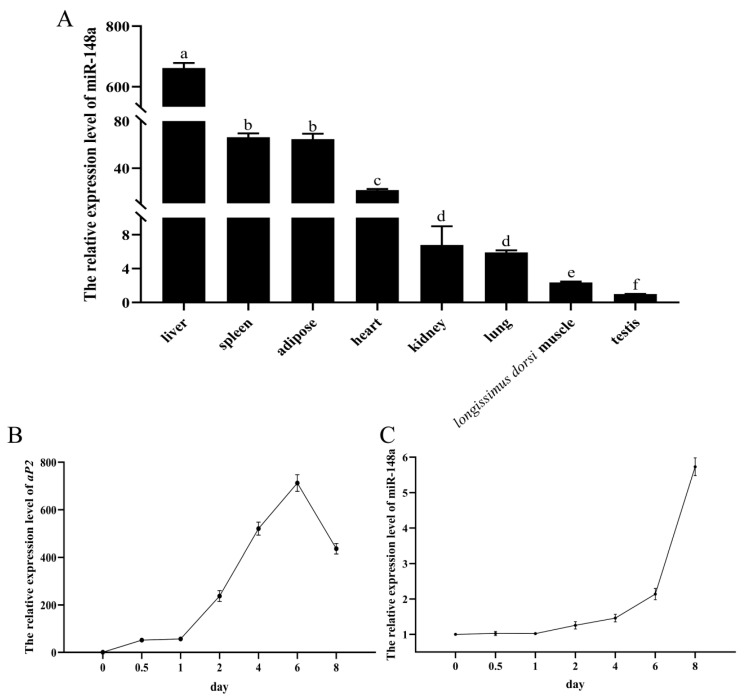
The tissue expression profiles of miR-148a and *aP2*. (**A**) The expression level of miR-148a in the eight different tissues of Tibetan ewes. The expression levels of *aP2* (**B**) and miR-148a (**C**) during differentiation of ovine preadipocytes using RT-qPCR. The results are depicted as the mean ± SD (*n* = 3). Means with a lower-case letter are significantly different (*p* < 0.05).

**Figure 2 animals-11-00820-f002:**
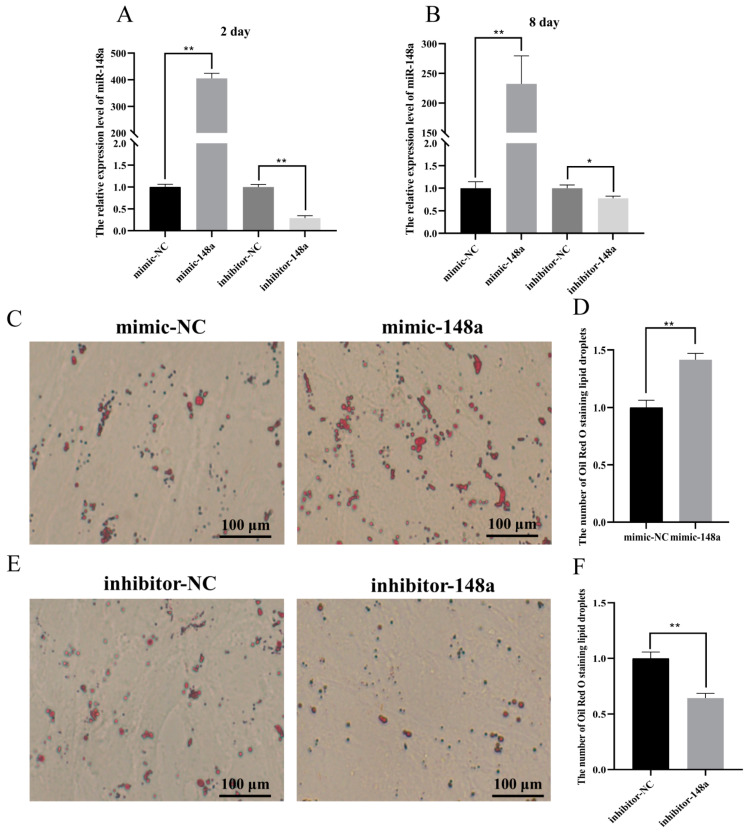
The miR-148a enhanced ovine preadipocyte differentiation. The miR-148a mimic, miR-148a inhibitor and negative control (NC) were transfected into preadipocytes at 50 nM. (**A**,**B**) The over-expression efficiency of miR-148a in adipocytes on day 2 and 8 after being transfected with the miR-148a mimic, miR-148a inhibitor and negative control (NC). (**C**) Oil Red O staining was performed in differentiated ovine preadipocytes after being transfected with miR-148a mimic and NC. (**E**) Oil Red O staining was performed in differentiated ovine preadipocytes after being transfected with miR-148a inhibitor and NC. (**D**,**F**) The areas of Oil Red O staining were counted using the ImageJ software. The results are depicted as the mean ± SD (*n* = 3), * *p* < 0.05, ** *p* < 0.01.

**Figure 3 animals-11-00820-f003:**
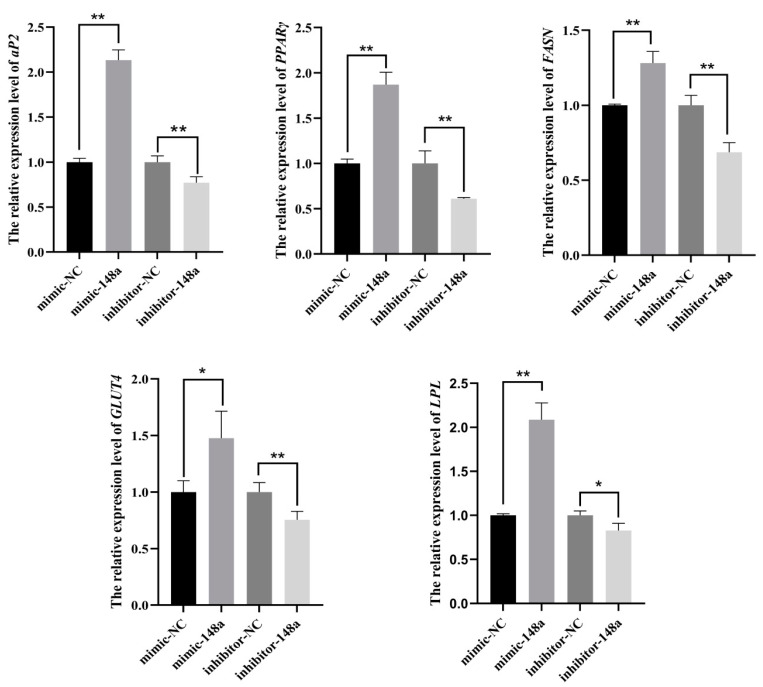
The miR-148a enhances preadipocyte differentiation. When miR-148a mimic, miR-148a inhibitor and negative control (NC) were transfected into differentiated preadipocytes at 50 nM, the expression levels of *aP2*, *FASN*, *PPARγ*, *GLUT4* and *LPL* were performed. The results are depicted as the mean ± SD (*n* = 3), * *p* < 0.05, ** *p* < 0.01.

**Figure 4 animals-11-00820-f004:**
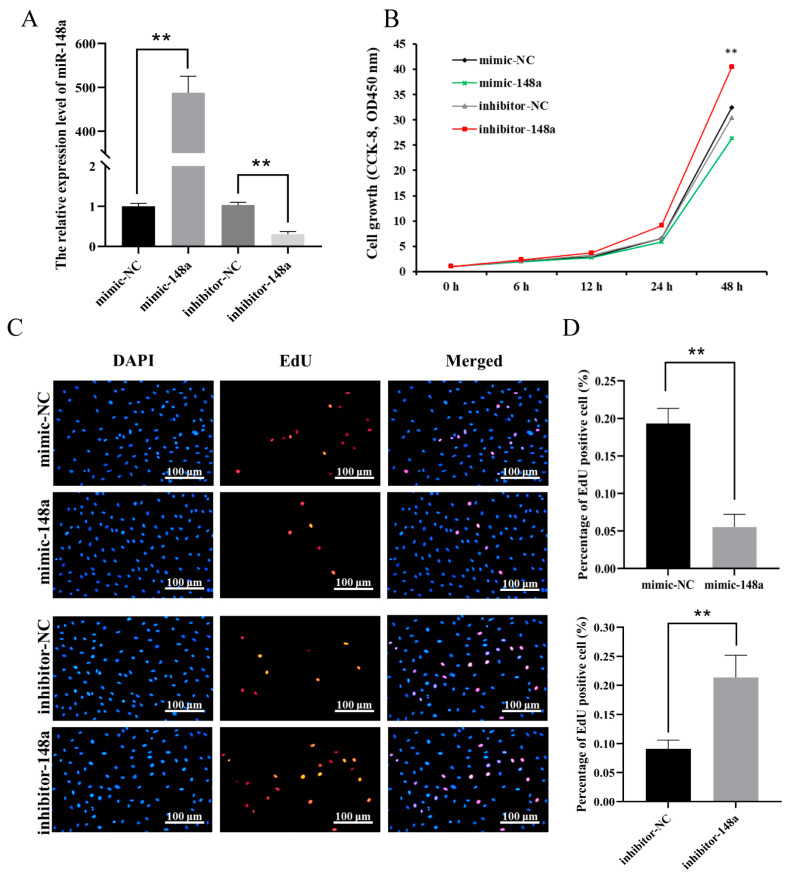
The miR-148a suppressed ovine preadipocyte proliferation. (**A**) The over-expression efficiency of miR-148a at 48 h in ovine preadipocytes after being transfected miR-148a mimic, miR-148a inhibitor and negative control (NC). (**B**) Effects of the miR-148a mimic, miR-148a inhibitor and NC on preadipocyte proliferation using CCK-8. (**C**) EdU was used to detect the effect of the miR-148a mimic, miR-148a inhibitor and NC on preadipocyte proliferation. The images of DAPI, EdU, and Merged groups reflect the total number of preadipocytes, the number of EdU-positive preadipocytes and the proportion of EdU-positive preadipocytes in the total preadipocytes, respectively. (**D**) The statistics of the number of EdU-positive preadipocytes. The results are depicted as the mean ± SD (*n* = 3), ** *p* < 0.01.

**Figure 5 animals-11-00820-f005:**
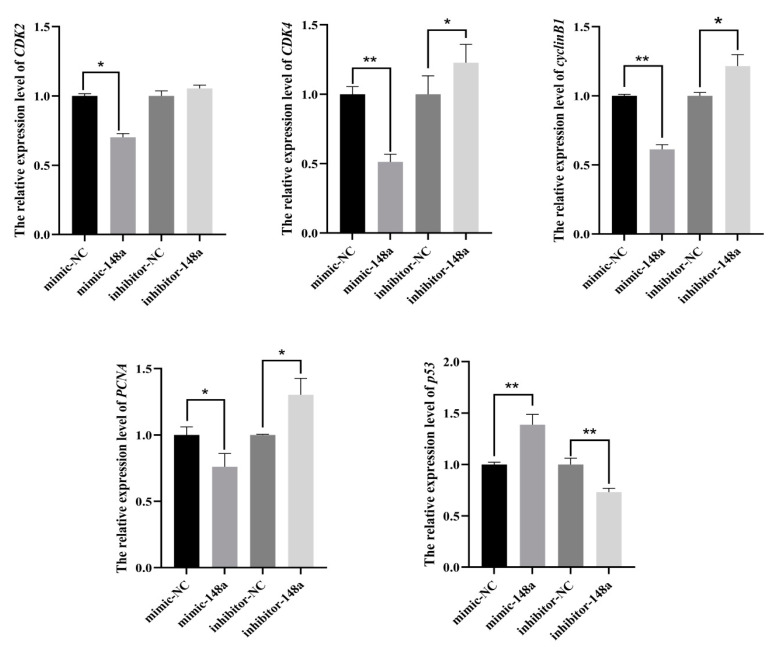
The miR-148a suppressed ovine preadipocyte proliferation. The expression levels of *CDK2*, *CDK4*, *cyclinB1*, *PCNA* and *p53* were detected when the miR-148a mimic, miR-148a inhibitor and NC were transfected into ovine preadipocytes. The results are depicted as the mean ± SD (*n* = 3), * *p* < 0.05, ** *p* < 0.01.

**Figure 6 animals-11-00820-f006:**
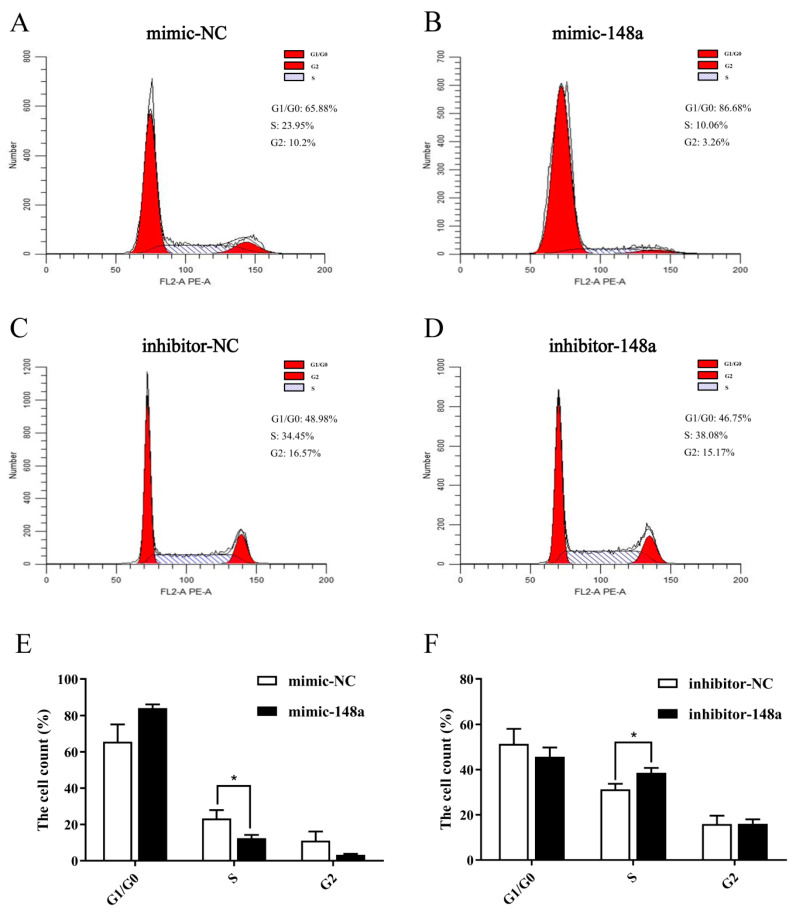
The miR-148a suppressed ovine preadipocyte proliferation. (**A**,**B**) The percentage of preadipocytes in different stages of mitosis was detected by flow cytometry after the preadipocytes were transfected with miR-148a mimic and NC. (**C**,**D**) The percentage of preadipocytes in each mitosis stage was detected by flow cytometry after the preadipocytes were transfected with the miR-148a inhibitor and NC. (**E**,**F**) The statistics of flow cytometry results. The results are depicted as the mean ± SD (*n* = 3), * *p* < 0.05.

**Figure 7 animals-11-00820-f007:**
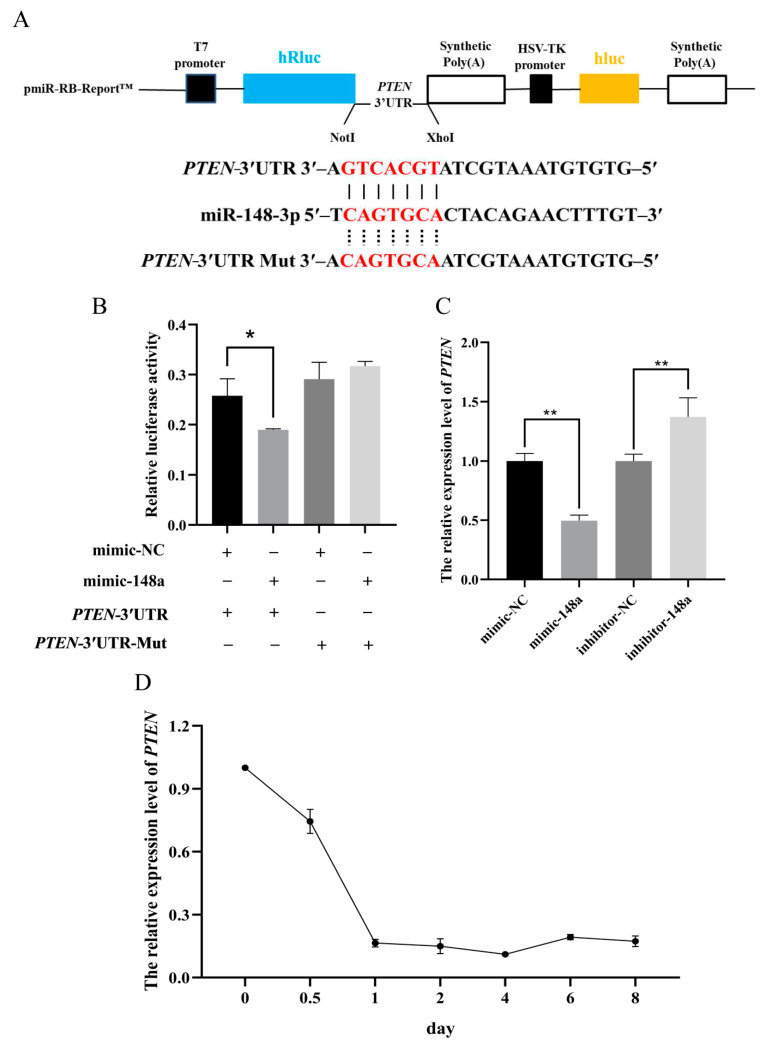
The miR-148a target *PTEN* in ovine preadipocytes. (**A**) The diagram of *PTEN*-3′UTR dual luciferase reporter vectors. (**B**) Dual luciferase assays were performed when miR-148a mimic, NC and wild-type or mutant-type *PTEN*-3′UTR report vectors were co-transfected into HEK293 cells. (**C**) The expression levels of *PTEN* in preadipocytes transfected with the miR-148a mimic, miR-148a inhibitor, and NC. (**D**) The expression profile of *PTEN* was detected during adipogenic differentiation. The results are depicted as the mean ± SD (*n* = 3), * *p* < 0.05, ** *p* < 0.01.

**Figure 8 animals-11-00820-f008:**
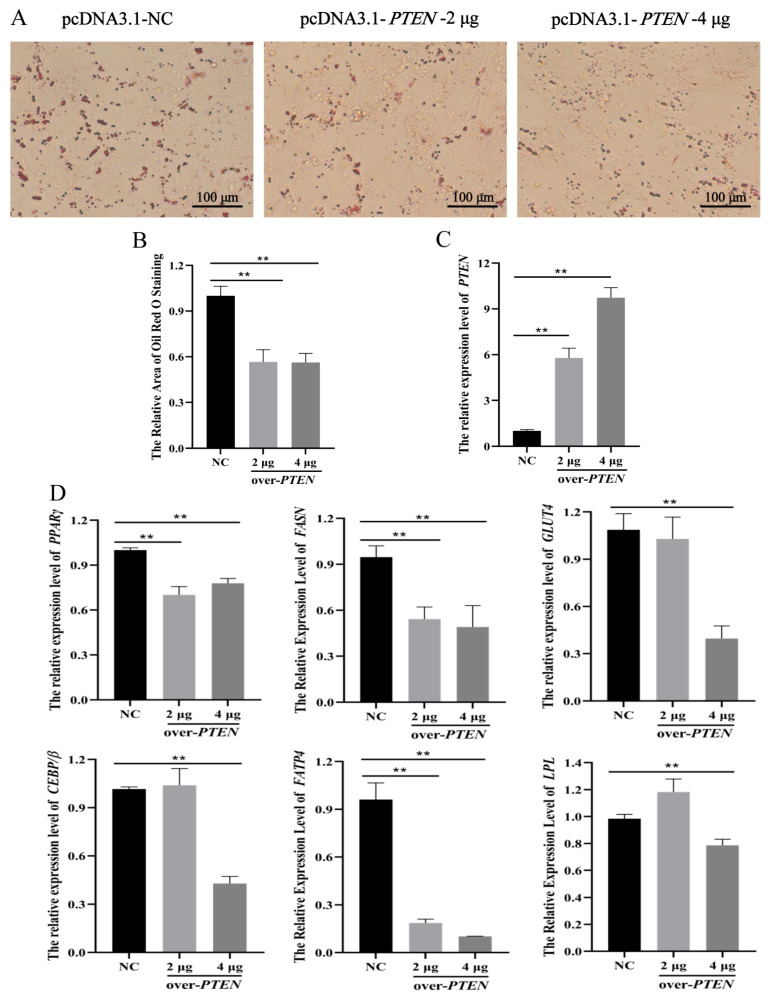
*PTEN* inhibited ovine preadipocyte differentiation. *PTEN* expression vector (pcDNA3.1-*PTEN*) and negative control (NC) were transfected into ovine preadipocytes. (**A**) Oil Red O staining was performed in differentiated ovine preadipocytes when 2 μg or 4 μg pcDNA3.1-*PTEN* expression plasmid were transfected into ovine preadipocytes. (**B**) The areas of Oil Red O staining were counted using the ImageJ software. (**C**,**D**) The expression levels of *PTEN* and adipogenesis marker genes were detected after the over-expression of *PTEN* in ovine preadipocytes. The results are depicted as the mean ± SD (*n* = 3), ** *p* < 0.01.

**Figure 9 animals-11-00820-f009:**
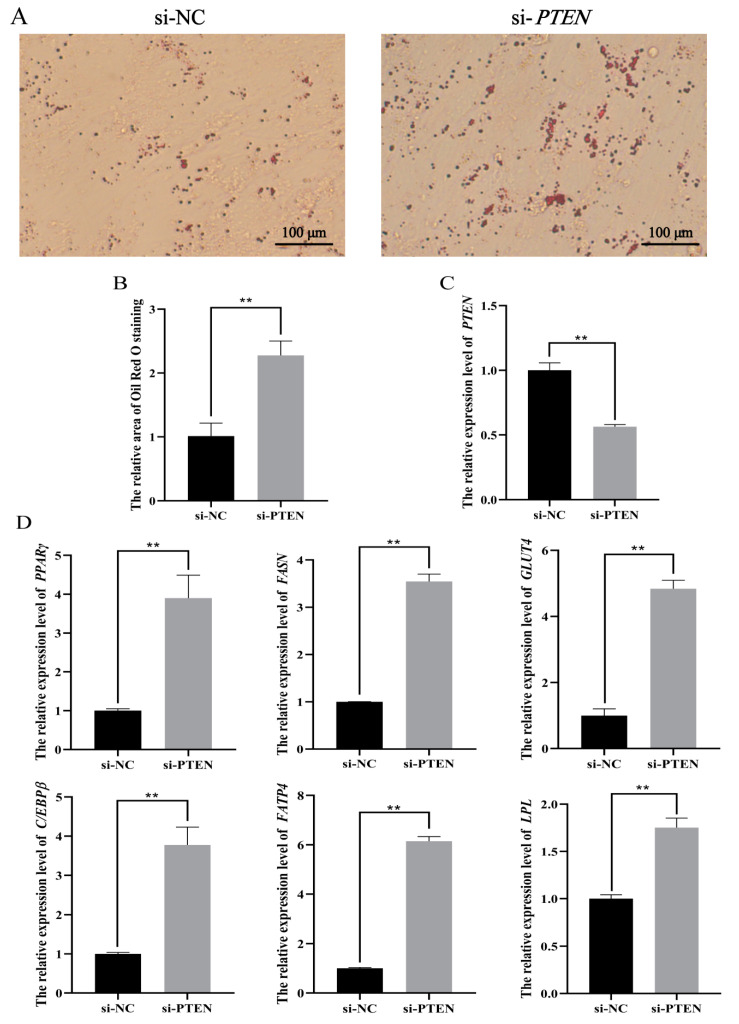
*PTEN* inhibited preadipocyte differentiation. si-*PTEN* or negative control (NC) were transfected into ovine preadipocytes. (**A**) Oil Red O staining was performed in differentiated ovine preadipocytes after *PTEN* silence. (**B**) The areas of Oil Red O staining were counted using the ImageJ software. (**C**,**D**) The expression levels of *PTEN* and adipogenesis marker genes were detected after transfection of *PTEN* siRNA in preadipocytes. The results are depicted as the mean ± SD (*n* = 3), ** *p* < 0.01.

## Data Availability

All relevant data is given in the paper. Additional information can be requested from the corresponding authors upon reasonable request.

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
