# Peer review of "MicroRNA-148a Regulates the Proliferation and Differentiation of Ovine Preadipocytes by Targeting PTEN"

_animals, 2021, doi:10.3390/ani11030820_

Round 1
Reviewer 1 Report
The corrections accepted.
Reviewer 2 Report
3rd March, 2021
Review of Manuscript ID animals-1143052, by Jin X. et al., entitled: “The microRNA-148a regulates proliferation and differentiation of ovine preadipocytes by targeting PTEN” that is intended for publication in Animals
(The Microsoft Word file as Reviewer Attachment for Manuscript ID animals-1143052 3rd March 2021 has also been added)
I recommend this manuscript for publication in Animals, provided that the minor revision will have been made by the Authors in the re-edited and resubmitted version of current paper according to the remarks of the Reviewer indicated below:
- The last names of the first authors have to be corrected in the References as follows (according to yellow-highlighted changes):
- Skrzyszowska, M.; Samiec, M. Enhancement of in vitro developmental outcome of cloned goat embryos after epigenetic modulation of somatic cell-inherited nuclear genome with trichostatin A. Ann. Anim. Sci. 2020, 20, 97–108.
- Olivera, R.; Moro, L.N.; Jordan, R.; Pallarols, N.; Guglielminetti, A.; Luzzani, C.; Miriuka, S.G.; Vichera, G. Bone marrow mesenchymal stem cells as nuclear donors improve viability and health of cloned horses. Stem Cells Cloning 2018, 11, 13–22.
- Samiec, M.; Romanek, J.; LipiÅ„ski, D.; Opiela, J. Expression of pluripotency-related genes is highly dependent on trichostatin A-assisted epigenomic modulation of porcine mesenchymal stem cells analysed for apoptosis and subsequently used for generating cloned embryos. Anim. Sci. J. 2019, 90, 1127–1141.
Author Response
Please see the attachment

This manuscript is a resubmission of an earlier submission. The following is a list of the peer review reports and author responses from that submission.
Round 1
Reviewer 1 Report
7th December, 2020
Review of Manuscript ID animals-1033128, by Jin X. et al., entitled: “The microRNA-148a regulates proliferation and differentiation of ovine preadipocytes by targeting PTEN” that is intended for publication in Animals
(The Microsoft Word file as Reviewer Attachment for Manuscript ID animals-1033128 7th December 2020 has also been added)
The present article has sought for the first time to investigate the effect of microRNA-148a on the processes of proliferating and adipogenically differentiating ovine preadipocytes. Moreover, the miRNA-148a-mediated genetic factor determining proliferation and adipogenic differentiation of preadipocytes in sheep have been detected for the first time. This factor encompasses a multifunctional tumor suppressor gene known as PTEN (phosphatase and tensin homolog deleted on chromosome ten; MMAC1, mutated in multiple advanced cancers 1).
An abundant methodological workshop based on the use of a variety of methods from the fields of molecular biology and cytobiochemistry should also be emphasized. Furthermore, it is noteworthy to mention that the Authors have used the relevant methods for statistical analyzing the results and have selected adequate references. This has allowed to thoroughly interpret and critically evaluate the results obtained by the Authors as compared to the results achieved by other investigators. Generally, the paper is very interesting, well written in English and provides many excellent-quality figures. The manuscript has been prepared in the format compatible with the requirements of Animals.
Nonetheless, I have several minor remarks and suggestions as follows:
Line 25 (on the page 1) …the expression of miRNA-148a were profiled…
has to be changed to either
….the expression levels of miRNA-148a were profiled….
or ….the expression of miRNA-148a was profiled….
Line 30 (on the page 1) …and had a highly expression…
has to be changed to either
…and had significantly increased expression….
or …and had significantly enhanced expression….
Additionally, I have one suggestion related to the Conlusions section as indicated below:
The Authors should add the following sentence at the end of this section, which, in my opinion, seems to considerably increase attractiveness of the results achieved by them:
For all the above-mentioned reasons, identification of molecular factors responsible for adipogenic differentiation and proliferation of ovine preadipocytes could increase practical usefulness and suitability of these cells for somatic cell cloning in sheep and other mammalian species as an alternative source of nuclear donors [48,49] in relation to foetal fibroblast cells [50,51], adult dermal fibroblast cells [52,53] and adult mesenchymal stem cells [54,55].
According to the above-mentioned sentence, the following 8 References should have been added:
- Tomii, R.; Kurome, M.; Ochiai, T.; Wako, N.; Ueda, H.; Hirakawa, K.; Kano, K.; Nagashima, H. Production of cloned pigs by nuclear transfer of preadipocytes established from adult mature adipocytes. Cloning Stem Cells 2005, 7, 279-88.
- Tomii, R.; Kurome, M.; Wako, N.; Ochiai, T.; Matsunari, H.; Kano, K.; Nagashima, H. Production of cloned pigs by nuclear transfer of preadipocytes following cell cycle synchronization by differentiation induction. J Reprod Dev 2009, 55, 121-7.
- Zhao, X.; Nie. J.; Tang, Y.; He, W.; Xiao, K.; Pang, C.; Liang, X.; Lu, Y.; Zhang, M. Generation of transgenic cloned buffalo embryos harboring the EGFP gene in the Y chromosome using CRISPR/Cas9-mediated targeted integration. Front Vet Sci 2020, 7, 199.
- Samiec, M.; Skrzyszowska, M. Preimplantation developmental capability of cloned pig embryos derived from different types of nuclear donor somatic cells. Ann Anim Sci 2010, 10, 385-398.
- Li, X.; Zhang, P.; Jiang, S.; Ding, B.; Zuo, X.; Li, Y.; Cao, Z.; Zhang, Y. Aging adult porcine fibroblasts can support nuclear transfer and transcription factor-mediated reprogramming. Anim Sci J 2018, 89, 289-297.
- Skrzyszowska, M.; Samiec, M. Enhancement of in vitro developmental outcome of cloned goat embryos after epigenetic modulation of somatic cell-inherited nuclear genome with trichostatin A. Ann Anim Sci 2020, 20, 97-108.
- Olivera, R.; Moro, L.N.; Jordan, R.; Pallarols, N.; Guglielminetti, A.; Luzzani, C.; Miriuka, S.G.; Vichera, G. Bone marrow mesenchymal stem cells as nuclear donors improve viability and health of cloned horses. Stem Cells Cloning 2018, 11, 13-22.
- Samiec, M.; Romanek, J.; Lipiński, D.; Opiela, J. Expression of pluripotency-related genes is highly dependent on trichostatin A-assisted epigenomic modulation of porcine mesenchymal stem cells analysed for apoptosis and subsequently used for generating cloned embryos. Anim Sci J 2019, 90, 1127-1141.
In conclusion, I strongly recommend this manuscript for publication in Animals, provided that the above-mentioned remarks and suggestions pointed out by the Reviewer will have been added by the Authors to the re-edited and resubmitted version of current paper.
Reviewer 2 Report
This is an interesting, well written paper, in a very “specific topic” in cell proliferation and differentation.
General comments
AKT functions as a cardinal nodal point for transducing extracellular (growth factors including insulin, IGF-1 and EGF ) and intracellular (such as mutated/activated receptor tyrosine kinases, PTEN, Ras and Src) signals.
It is inhibited by phosphatase PTEN. Deregulation of the PI3K/PTEN/AKT pathway is one of the most common altered pathways in human malignancy.
In the past few years, significant advances have been made in the understanding of AKT signaling in human oncogenesis and the development of small molecule inhibitor of AKT pathway. The key roles of the RAS/RAF/MEK/ERK, PI3K/PTEN/AKT/mTORC1, TP53 microRNAs (miRs) as therapeutic targets are discussed and targeting the PTEN gene and/or protein will likely provide an efficient strategy for therapeutic intervention in cancer and metabolic diseases like type 2 diabetes mellitus, obesity, and cardiovascular dysfunction.
This study provides new insights that miR-148a inhibits ovine adipose cell proliferation, and accelerates differentiation by negative regulation of the PTEN/AKT pathway.
Reviewer 3 Report
Overall research and writing quality are very good. Authors created a set of valuable information to be contributed in the relevant literature. Therefore, it is strongly suggested for publication after a few minor corrections and suggestions to be considered which are outlined below.
- In general, the language of the paper is of excellent quality. However, a proofreading is strongly suggested, since there are tense inconsistencies, vocabulary and collocation errors in sentences bearing significant information. This mildly creates problems for fully comprehend what was meant by those sentences and interferes with the coherency of this valuable piece of work. Some of the problematic sentences include the lines 70, 131, 157-158, 200, 369-371, 383 and so on.
- Regarding methodology, especially for characterizing expression levels of different sheep tissue samples, it is important to note that only samples from 4 sheep with unknown kinship were used. Certain results may have appeared due to the covariance between relatives, animal’s genetic background and/or low number of animals used in the study. Therefore, I would suggest to avoid strong statements such as 235-236 and rephrase it as a deduction.
- In figure 2B, expression level of aP2 appears to start from zero. I would suggest to re-visit the figure and check if that is conceptually correct.
- In line 363, there is an expression of “miR-148a inhibited cells from mitotic G1 to S phase”. I assume it is meant to be “miR-148a inhibited cells moving from mitotic G1 to S phase”. Again in lines 370-371, the words “level” are redundant.
- Finally, I would suggest to revisit the first sentence of conclusion as the structural problems create loss of meaning.
Reviewer 4 Report
Dear Authors,
The study is interesting and comprehensive, however, the manuscript, especially the methods section, is chaotic in some parts and not described enough. Information on some analyses and genes is missing. Moreover, your conclusions are not the same in different parts of the manuscript, so overall, it is not clear what your study showed eventually. More detailed information on Ethics approval is also needed. Detailed comments are in the text.
